# An AI-Enabled Approach in Analyzing Media Data: An Example from Data on COVID-19 News Coverage in Vietnam

**Quan-Hoang Vuong** [1], **Viet-Phuong La** [1], **Thanh-Huyen T. Nguyen** [1], **Minh-Hoang Nguyen** [1], **Tam-Tri Le** [1,2] **and Manh-Toan Ho** [1,*]

[1]   Centre for Interdisciplinary Social Research, Phenikaa University, Ha Noi 100803, Vietnam;
       hoang.vuongquan@phenikaa-uni.edu.vn (Q.-H.V.); phuong.laviet@phenikaa-uni.edu.vn (V.-P.L.);
       huyen.nguyenthanhthanh@phenikaa-uni.edu.vn (T.-H.T.N.);
       hoang.nguyenminh@phenikaa-uni.edu.vn (M.-H.N.); letamtri10@gmail.com (T.-T.L.)
[2]   AI for Social Data Lab, Vuong & Associates, Ha Noi 100000, Vietnam
[*]   Correspondence: toan.homanh@phenikaa-uni.edu.vn

**Abstract:** This method article presents the nuts and bolts of an AI-enabled approach to extracting and analyzing social media data. The method is based on our previous rapidly cited COVID-19 research publication, working on a dataset of more than 14,000 news articles from Vietnamese newspapers, to provide a comprehensive picture of how Vietnam has been responding to this unprecedented pandemic. This same method is behind our IUCN-supported research regarding the social aspects of environmental protection missions, now appearing in print in Wiley's *Corporate Social Responsibility and Environmental Management*. Homemade AI-enabled software was the backbone of the study. The software has provided a fast and automatic approach in collecting and analyzing social data. Moreover, the tool also allows manually sorting the data, AI-generated word tokenizing in the Vietnamese language, and powerful visualization. The method hopes to provide an effective but low-cost method for social scientists to gather a massive amount of data and analyze them in a short amount of time.

**Dataset:** https://zenodo.org/record/4738602

**Dataset License:** Creative Commons Attribution 4.0 International

**Keywords:** online media data; content analysis; AI-enabled web crawling

## 1. Summary

Newspapers are an essential source for social researchers to learn about different aspects of the ecology and society, such as climate change [1], the pandemic [2], corporate social responsibility [3], or environmental degradation [4,5]. Qualitative content analysis is one of the most common methods to comprehensively study the content of the media [6]. However, with the development of the Internet, the daily high publishing volume from newspapers has made the traditional way of content analysis harder. Due to this reason, research switched to tools that could automatically scan news outlets and collect news articles to save time and cost [7]. For example, Brandy and Diakopoulos created Python programs to curate stories and the "share" links of each news article before storing them in a CSV file. They did not collect the content of these articles because the purpose of their study was only to compare the degree of personalization between human-curated and algorithmically curated sections [8]. There are studies that manually collected news before automatically analyzing the content. A study conducted in China that collected news in the China National Knowledge Infrastructure (CNKI) database using a set of key-

words related to COVID-19 and tourism retrieved 794 articles published in a month, before yielding only 499 articles for analysis [9]. This task would be burdensome in the case of extending the period of publishing and number of news outlets and websites, as we would likely obtain thousands of articles. This study did use an automated program to tokenize news articles and extract the top keywords. However, no detail concerning how accurately the program used in this study processed the language was provided. On this matter, despite there being toolkits that were successfully applied to English documents [10], studies about Vietnamese documents had a concern as to whether the techniques originally used on English documents could be applied to Vietnamese documents because these languages have many differences in processing [11–13]. Researchers proposed using Bag of Words with keyword extraction and Neural Network to solve the problem related to Vietnamese news classification [11].

During the outbreak of COVID-19 in Vietnam in March 2020, our research team has decided to study the COVID-prevention policy from the government, the reaction from the media, and the scientific community. Due to the number of websites and the long period featured in the study, the number of articles was expected to be too much to obtain manually. Furthermore, the concern regarding the ability of a machine to process the content of the articles was prevailing. In this light, we needed a tool that can (1) scan Vietnamese news outlets and official government websites to identify news articles with a set of keywords, download their raw HTML data and extract unrelated content; (2) remove duplicated articles (similarity over 90%); and (3) process Vietnamese effectively. However, there is no tool that is suitable for acquiring the data to satisfy our needs. Available tools can brilliantly perform the first and second steps but not the third, due to the difficulties to split Vietnamese words and phrases.

Difficulties to process Vietnamese words and phrases correctly with machines have been well documented [11–14]. While tokens of languages such as English and Russian are words, Vietnamese has sub-syllables, syllables and words [15], so using homogeneous unit systems could lead to incorrect splitting and disastrous misunderstanding. Furthermore, the Vietnamese language is complex, so it is easy to split words and phrases in the wrong way. Thus, tokenizing Vietnamese phrases should be performed carefully based on the context to achieve the best results. In addition, Vietnamese has words that are written in the same way but have different meanings. This could lead to misleading analyses. For example, the word "cơ quan" could mean "body organs" or "a government agency", depending on the context. When tool lacks the ability to extract the contextual information, which is usually hidden and not extractable [16,17], the tool could misunderstand the meaning of a word even when a sentence has been split correctly, and this might miscount the frequency that the word appears, which lead to a wrong analysis. As the COVID-19 pandemic influenced a broad range of fields, there might be variation in meaning across different contexts. Thus, there are words that require careful definition before being used in other fields to avoid misunderstanding [18]. For these reasons, we needed to customize our programs, which is a homemade AI-enabled software using Python codes, to collect news from online sources and process Vietnamese better. Effective use of an AI module was suggested to be the solution to problems of information retrieval and analysis [19].

Eventually, a dataset of 14,952 news articles from 14 online news outlets was collected and stored in the software's web-based database using the .NET core, leading to a successful publication [2]. Later on, an early version of the software also contributed to another publication about the moral–practical gaps in corporate social responsibility (CSR) missions of Vietnamese firms [3].

This article provides a comprehensive introduction to our news curation method. In the next section, the overview of the system is presented. Then, we show the workflow of the web crawler, following by the logic behind our keywords and the user interface of the web-based database. In the conclusion section, the contribution, limitations, and future research direction of the system are discussed.

**2. Data Description**

The dataset extracted from the AI-enabled news crawler contains the following information:

- Date: The date of publication of the crawled news articles.
- Title: The title of the crawled news articles.
- Url: The Uniform Resource Locators (URLs), or the web addresses, of the crawled news articles.
- Detail: The content of the crawled news articles, which will be used for analysis.

Based on the basic information, the system will analyze the dataset using keywords or text analyzers. The detail will be described in the Methods section.

**3. Methods**

*3.1. System Overview*

The system has five main components:

- Projects and Data Sources: The component includes online websites allocated into different categories, the HTML parse of the websites, projects, and sources of data for each project, as well as the projects' data filter.
- Data Logging: The component stores the queue of crawling URLs and notes on the collected or appropriate news.
- News and Keyword Filter: The component includes structured news articles that were collected and filtered according to keywords for each project.
- Research Subjects and Characteristics: Tables contain information about the research subjects, their characteristics, connection among research objects, and news articles.
- Text Data Analyzer: Tables consist of a Vietnamese dictionary, a data table of place names and proper names, and a data table to train the artificial intelligence (AI) model in separating Vietnamese words.

The components and the data tables are depicted in Figure 1.

*3.2. Crawling Workflow*

News Crawler is a module coded in Python with the Scrapy Framework to execute the News Crawling and Content Extraction. These orders have to satisfy the following criteria: (1) collecting data from a configured website; (2) extracting a news article's content and metadata; and (3) easily configuring a new news source. In Figure 2, the workflow of the tool is visualized as follows:

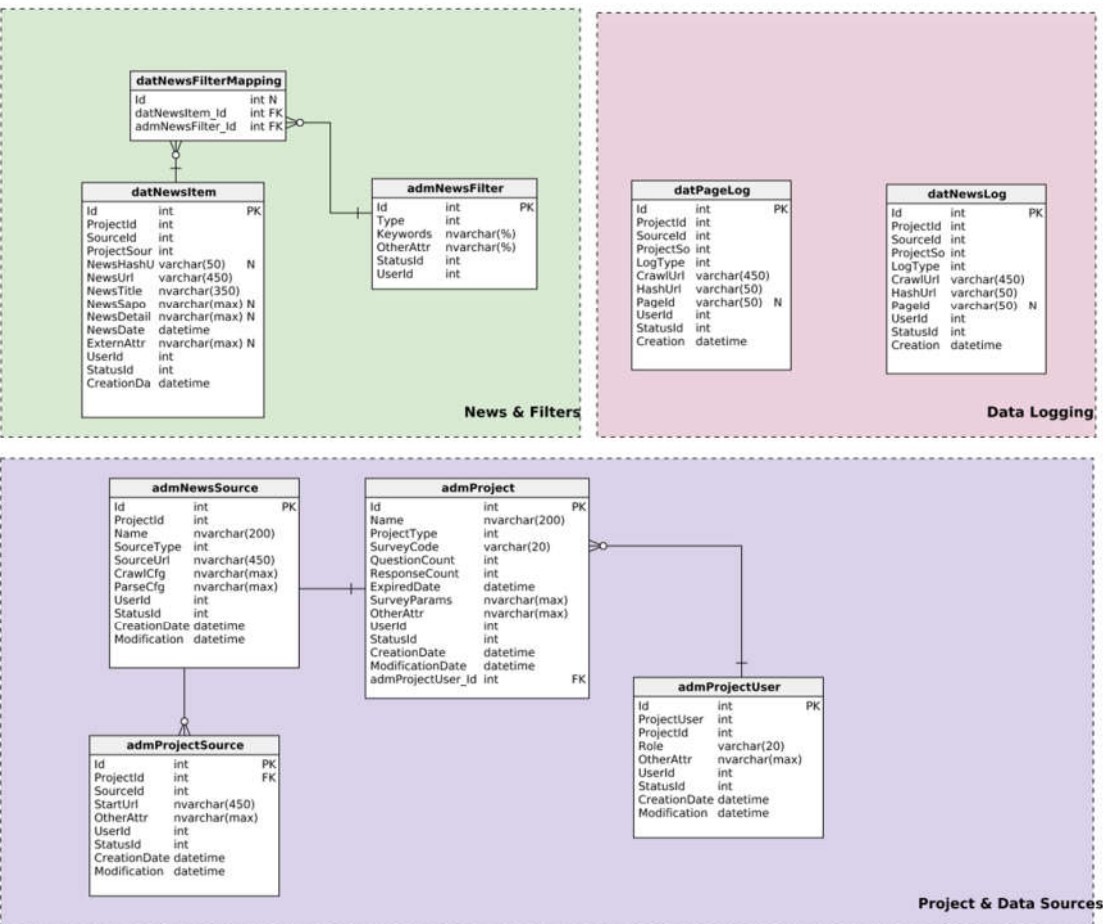

**Figure 1.** An overview of the system.

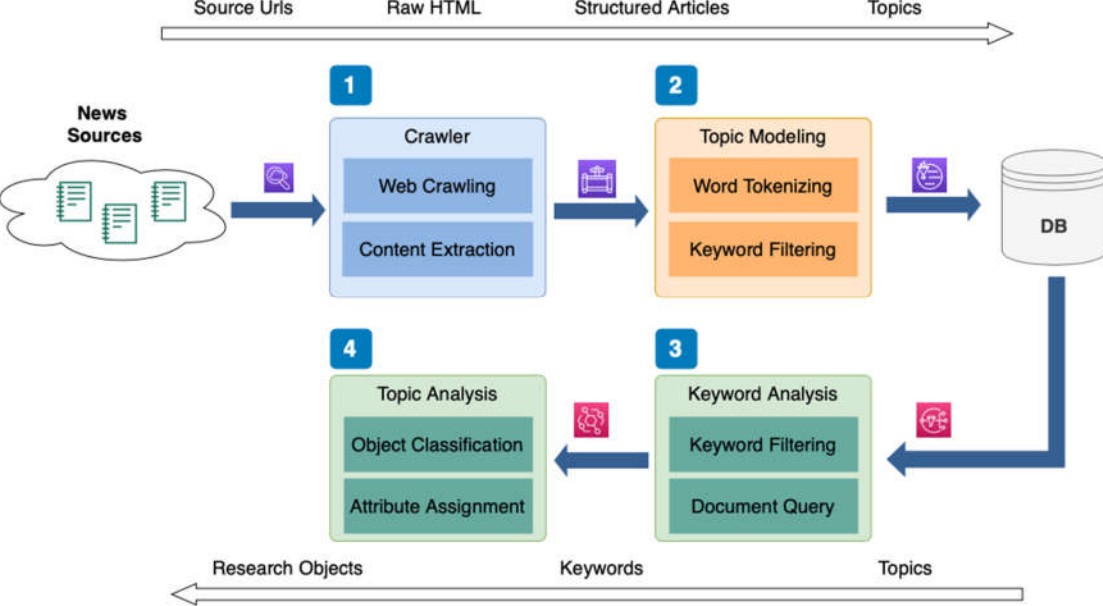

**Figure 2.** The workflow of News Crawler.

*3.3. Crawler*

From the news sources, the Crawler module will execute Web Crawling and Content Extraction. Web Crawling has three sub-tasks: (1) scan the websites and find news articles by using the "Scrapy Framework"; (2) download raw HTML data of each news article; and (3) recursively scan the following websites or internal links that satisfy the criteria and repeat Step (1).

News websites have different page structure; thus, the Crawler collect articles from a page in the queue and then continue with the next page. The recursive function stops when there are no more websites meeting the criteria, or all the news articles are collected. An activity log of the whole process is recorded to ensure no article is collected twice.

After the Web Crawling, Content Extraction extracts the content and metadata of the news article, such as the title, lead paragraph, main content, author, or date, from the raw HTML data. The module ensures unnecessary information, such as a menu or advertisement, is left out. The extraction function is based on HTML parsing to extract the information. Moreover, due to the difference in the structures of the news websites, the metadata is formatted differently, and some websites even have an advertisement frame in-between the news article. Therefore, the extraction of content has to be configured based on each news source. The outcome is a set of structured articles with the main content stored in plain text format, and the metadata is saved accordingly. The structured articles are checked for duplication by using the Duplicate Finder module.

It is common for many newspapers to report one story. Thus, Duplicate Finder uses Levenshtein distance to compare the title and the content of the structured news articles and then flags the duplicate articles. When the similarity among articles is up to 90%, they are flagged as duplication. A news article with many duplications suggests many sources have reposted the article. This is useful information to identify the importance and impact of an article.

Then, Topic Modelling (No. 2, Figure 2) will use Keyword Filtering and Word Tokenizer to filter the structured articles. Structured articles that have been Word Tokenized will have a bag-of-words model in their data. Keyword Filtering searches for a set of keywords that was structured in a logical tree format in an article. The filter is used in extracting the articles based on research topics, extracting keywords based on research objects, or thoroughly analyzing the density of keywords in a project's news articles.

*3.4. Word Tokenizer*

Word Tokenizer is an AI tool that separates words and phrases in the Vietnamese language. In multi-syllable languages, such as Vietnamese, the separation of words is essential for searching and analyzing keywords, or more commonly, natural language processing. Currently, developers have produced good programs for popular languages such as English. Notable examples include Natural Language Toolkit (https://www.nltk.org/ (accessed on 25 June 2021)) or spacy (https://spacy.io/api/tokenizer (accessed on 25 June 2021)). However, there are not many similar programs for Vietnamese, and for the available options, the features are limited [7,11,12]. Due to these limitations, we were unable to control the accuracy, and we also had to retrain the model to fit our research topics. Thus, we developed a homemade AI-enabled Word Tokenizer for the Vietnamese language, using the Conditional Random Fields (CRFs) algorithm—which is based on the *sklearn_crfsuite* library. The data to support the model includes the Vietnamese dictionary (70,000 words), 671 place names, and 14,000 proper names in Vietnamese.

For instance, a multi-syllable word in Vietnamese, such as "xét nghiệm" (test), will be defined by the first syllable, "xét". Then, the word will be tagged with BW. The following syllable will be tagged IW. With the following original sentence:

Bộ Y tế chiều 28/8 ghi nhận hai ca nhiễm nCoV, trong đó một ca Đà Nẵng, một ca rời khu cách ly tại Hải Dương về đến Hà Nội thì xét nghiệm dương tính.

We will have the corresponding tag:

[BW] [BW IW] [BW] [BW] [BW IW] [BW] [BW] [BW] [BW] [O] [BW IW] [BW] [BW] [BW IW] [O] [BW] [BW] [BW] [BW IW] [BW] [BW IW] [BW] [BW] [BW IW] [BW] [BW IW] [BW IW]

Thus, the sentence after going through the Word Tokenizer is as follow:

[Bộ] [Y tế] [chiều] [28/8] [ghi nhận] [hai] [ca] [nhiễm] [nCoV], [trong đó] [một] [ca] [Đà Nẵng], [một] [ca] [rời] [khu] [cách ly] [tại] [Hải Dương] [về] [đến] [Hà Nội] [thì] [xét nghiệm] [dương tính]

A sentence will be separated into chains of syllables. The Word Tokenizer will identify and tag the syllable. The criteria to build the AI feature include recognizing syllables as BW or IW or O (not-a-word), and the feature was trained by using the database of the tagged news articles. The criteria to identify a syllable in a chain are as follow:

- A syllable joined with 2 syllables after it to create 3 syllables (tri-gram) in the dictionary.
- A syllable joined with 2 syllables in front of it to create 3 syllables (tri-gram) in the dictionary.
- A syllable joined with 2 syllables in front of and after it to create 3 syllables (tri-gram) in the dictionary.
- A syllable joined with a syllable after it to create 2 syllables (bi-gram) in the dictionary.
- A syllable joined with a syllable in front of it to create 2 syllables (bi-gram) in the dictionary.
- A syllable that is at the beginning of a sentence.
- A syllable that is at the end of a sentence.
- A syllable that is a number.
- A syllable that contains a number.
- A syllable that is a special character.

Examples of the code used are presented in Table 1:

**Table 1.** Examples of the code for developing the Word Tokenizer.

| Purposes | Code |
|---|---|
| **Building the features** | features = { 'w.islower': word.islower(), 'w.isupper': word.isupper(), 'w.istitle': word.istitle(), 'w.ispunct': word.isPuncts(), 'w.isBOS': isBOS(), 'w.isEOS': isEOS(), '-1:w.bi_gram': ' '.join([word1, word]).lower() in bi_grams, '+1:w.bi_gram': ' '.join([word, word1]).lower() in bi_grams '-2:w.tri_gram': ' '.join([word2, word1, word]).lower() in tri_grams, '+2:w.tri_gram': ' '.join([word, word1, word2]).lower() in tri_grams, } |
| **Training the model** | model = sklearn_crfsuite.CRF( algorithm='lbfgs', c1=0.1, c2=0.1, max_iterations=2000, all_possible_transitions=True, model_filename='models/model.bin' ) model.fit(X_train, y_train) y_pred = model.predict(X) |

### 3.5. Analysis

Based on the requirement of the analysis, the news articles in the database can be filtered or queried accordingly by using Keyword Analysis (No. 3, Figure 2). The analysis will show the number of keywords through time and build a keyword map.

The Topic Analysis (No. 4, Figure 2) allows the creation of research objects. A news article will be filtered by keywords to check whether it mentions the research objects or not. A suitable article will be tagged with a research object or more. A research object will have many characteristics. The values of these characteristics are configured based on the news articles that mentioned the research object. The characteristics can be added manually or automatically. For example:

- Research object: Environment-related events, organizations participated in the events.
- Characteristics of the research object: Type of event, time of the event, the degree of impact, type of organization.

### 3.6. Keywords Logical Tree

The keywords are structured into a logical tree, which allows easy and effective configuration of the keyword filter. A keyword group is a group of keywords that are connected by a logical-mathematical operator (AND, OR, AND NOT). A news article meets the criteria of a keyword group when it satisfies all of the keywords and the logical-mathematical operator in the group.

For example, when we studied the news about COVID-19 in Vietnam, we can have a group of three keywords: covid, coronavirus (or corona virus), sars-ncov-2, with the logical-mathematical operator AND. In this case, a news article meets the requirements when it contains all three keywords in the group (covid AND corona virus AND sars-ncov-2).

In a similar group of keywords, but with the logical-mathematical operator OR, a news article is appropriate when it has at least one keyword (covid OR corona virus OR sars-ncov-2).

In the case of the logical-mathematical operator AND NOT, a news article is satisfied with the condition when it does not have any keyword in the group (NOT covid AND NOT corona virus AND NOT sars-ncov-2).

Finally, each group can contain each other, and a son group will have a role as a keyword in the father group. Using this method, we can execute logical combinations with higher complexity; for instance, (covid OR corona virus OR sars-ncov-2) AND (policies OR government OR lockdown).

### 3.7. Object Classification

Object classification is tagging news articles based on the research criteria. For instance, in the COVID-19 articles [2], we wanted to evaluate the responses of the Vietnamese government. Thus, we have identified and classified news articles into different groups that related to government's response: fake news prevention, education, market control, or social distancing. Eventually, we were able to identify different types of response in different time period.

The classification module is a Text Classification AI that was built based on the *sklearn* library. The module uses the *naive_bayes* algorithm, which is based on the keywords of each article. We used the abovementioned Word Tokenizer to generate a document term matrix for the model. At the beginning, we classified the objective manually. When the objective contains an adequate number of samples, we used the classified objective to train the AI. Examples of the code used for training the AI are as follow:

```
vectorizer = CountVectorizer(lowercase=True,ngram_range = (1,1),tokenizer =
WordTokenizer.tokenize)
X = vectorizer.fit_transform(x)

X_train, X_test, y_train, y_test = train_test_split(
X, y, test_size=0.3, random_state=1)

# Model Generation Using Multinomial Naive Bayes
clf = MultinomialNB().fit(X_train, y_train)
predicted= clf.predict(X_test)
```

*3.8. User Interface*

A Web Application on the .NET Core platform was built to make the core functions such as creating a project, configuring news sources, extracting data, searching, analyzing, and visualizing keywords. The Web Application is accessible from the URL: http://na.aisdl.com/ (accessed on 25 June 2021).

Project Management and Configuration

Figure 3 presents how a news source is configured. An online newspaper named VnExpress (URL: https://vnexpress.net/ (accessed on 25 June 2021)) was set up as a news source for the News Crawler. We specifically targeted the business section of the newspaper; thus, the URL is https://vnexpress.net/kinh-doanh/p{0} (accessed on 25 June 2021).

Tên nguồn tin

vnexpress .net

Loại nguồn tin

Normal

Url (mặc định)

https://vnexpress.net/kinh-doanh/p{0}

📋 Cấu hình lấy trang tin

NextPageType

Number ▾

NextPageFormat

Enter the value for NextPa

NextPageStart

1 ⊗

NextPageStep

**Figure 3.** News sources configuration.

Figure 4 shows the configuration for a project. The project Covid-19 has seven keywords (ncov, sars-cov, corona, covid, viêm phổi lạ, virus viêm phổi, 2019n-cov) with the mathematical operator OR.

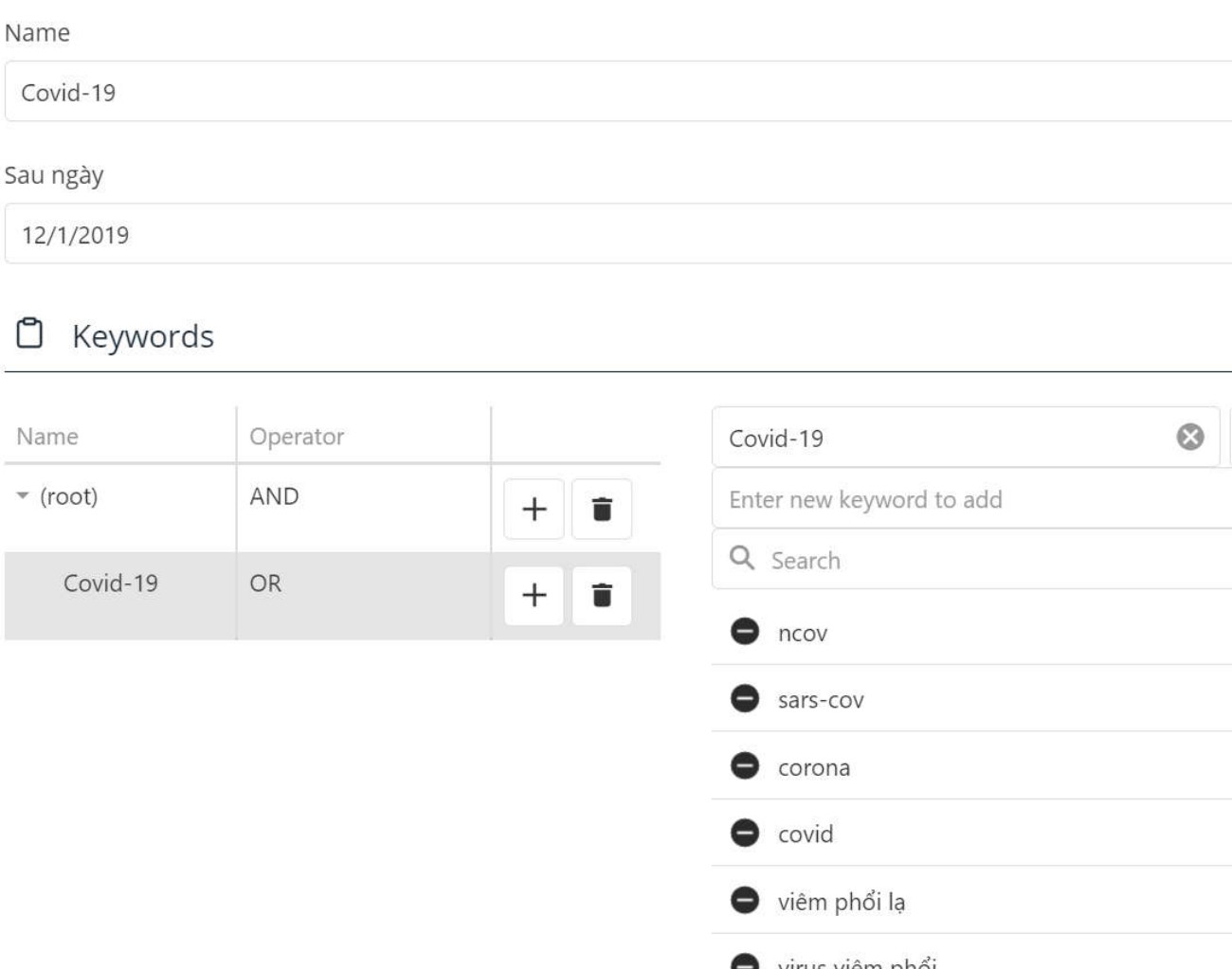

**Figure 4.** Configuration for a project.

### 4. Examples of Analysis

We use the *d3.js* library (https://d3js.org (accessed on 25 June 2021)) to visualize the research results. The crawled news articles were stored according to its published time. Therefore, we were able to analyze the data in different time periods.

For example, when the crawling process is done, we have the results as provided in Figures 5 and 6. In Figure 5, the system provides a summary of the number of articles from each source, and the number of articles through time. The system provides a list of news articles to examine the data more closely, as in Figure 6. Researchers can quickly read the title and summary and access the full article via the provided URL. Moreover, the system also allows the researchers to manually delete an article or to extract the list to CSV format.

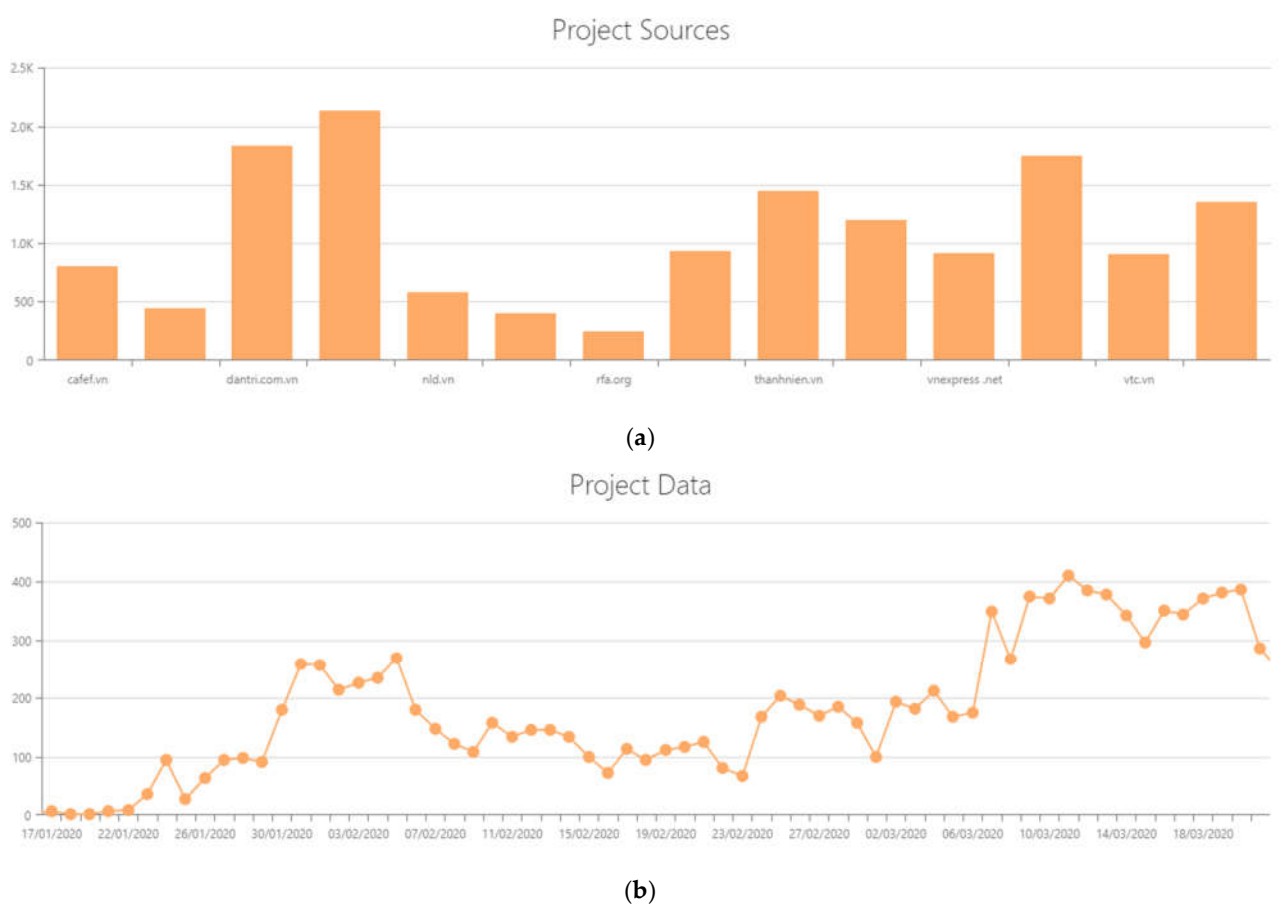

**Figure 5.** News crawler results: (**a**) number of crawled articles by sources; (**b**) number of crawled articles by date.

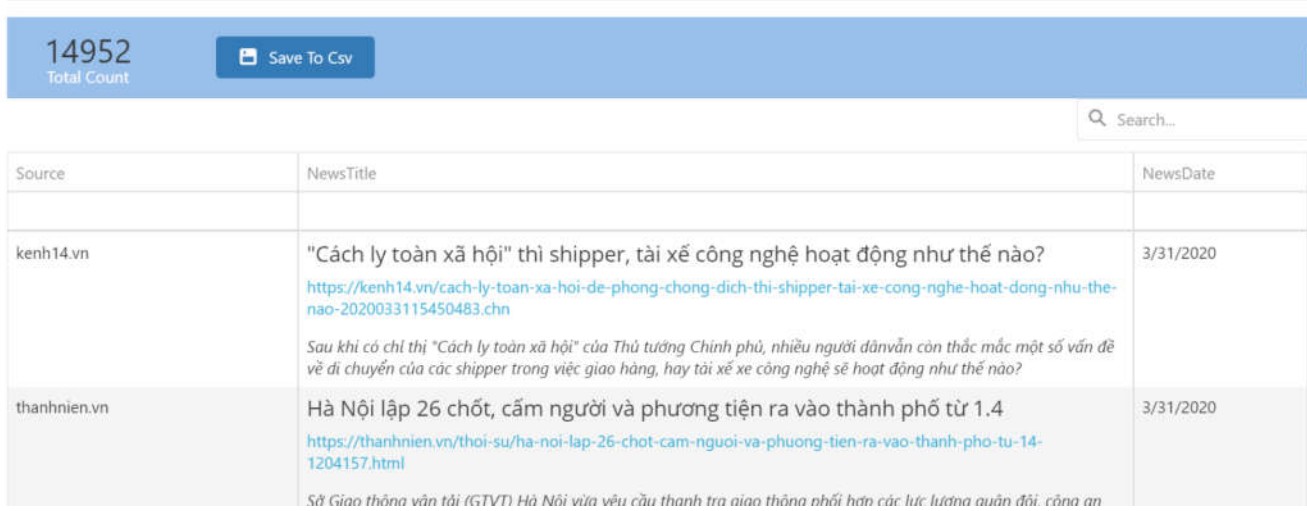

**Figure 6.** List of news articles.

After cleaning the data manually, the researchers can start exploring the dataset via the Keyword Filtering tool. Figure 7 presents four topics that we explored in the COVID-19 project [2]. We have different sets of keywords for Policy, Media, Science, and Socioeconomics.

| 4 Total Count | | | |
|---|---|---|---|
| Id | Name | FilterCfg | LastUpdate |
| 1 | Policy | ["chính phủ","thủ tướng","bộ y tế","sở y tế","phòng ngừa","biện pháp","cách ly","kiểm tra","hỗ trợ"] | 3/21/2020 |
| 2 | Media | ["mạng xã hội","cư dân mạng","facebook","zalo"] | 3/31/2020 |
| 4 | Science | ["khoa học","nghiên cứu","lây nhiễm cộng đồng"] | 3/21/2020 |
| 5 | Socioeconomics | ["kinh tế","bình ổn","quan hệ quốc tế","biến động"] | 8/30/2020 |

**Figure 7.** Keyword filtering for analysis.

The system provides statistics for each topic, both by the number of news articles and the number of news articles for each keyword (see Figure 8). Similar to the list of news articles in Figure 6, the analysis of topics also provides access to the list of articles for each keyword. The system also generates a word cloud for data visualization (See Figure 9).

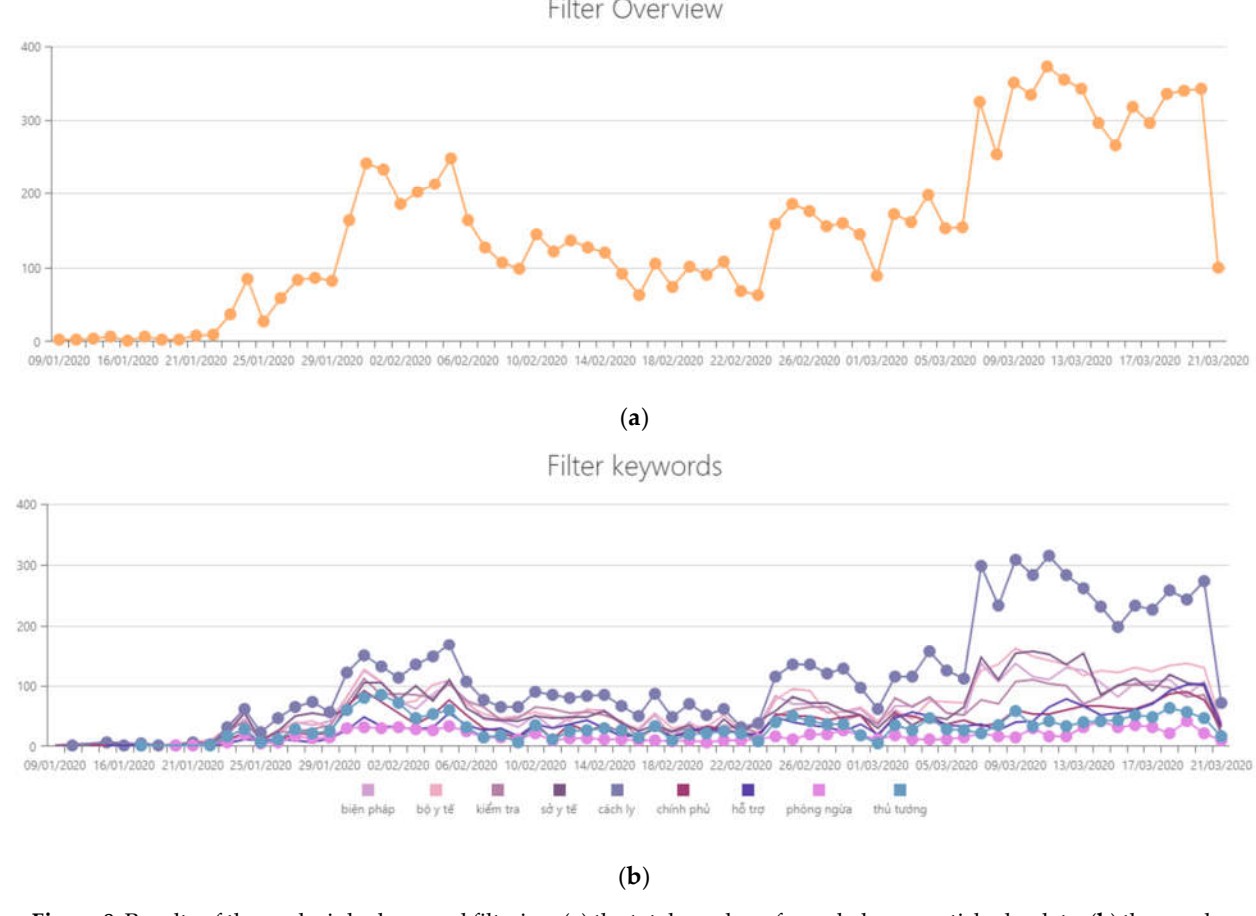

(**a**)

(**b**)

**Figure 8.** Results of the analysis by keyword filtering: (**a**) the total number of crawled news articles by date; (**b**) the number of crawled articles by date and keywords.

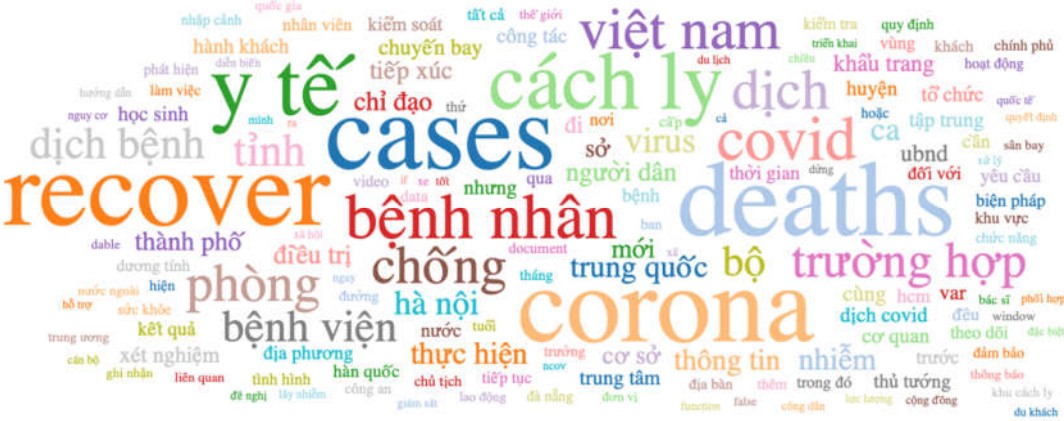

**Figure 9.** Word cloud generated by the system.

## 5. Limitations

However, there are still limitations [20]. Firstly, the News Crawler has to rely on the archive of the news sources to scan the news articles. If a news source does not archive their articles for a long time, the News Crawler will not be able to obtain adequate data. Similarly, several news sources may also have had the kind of web layout that hinders the ability to flip a page and automatically scan for data. Therefore, users need to choose the news sources carefully to make the most out of the news analysis. Secondly, the AI needs a large amount of data to learn from. At the current stage, the Word Tokenizer needs to learn more to perform flawlessly. Lastly, the system is currently focused on newspapers and official media outlets. In the future, we hope to focus on social media, where the psyche of the society is reflected. Nevertheless, a news analysis has potential in helping social scientists, mostly from Vietnam, to extract and learn from the online media data.

## 6. Usage Notes

The dataset provided more than 14,000 news articles about COVID-19, and it was quickly analyzed in just a week to provide insight into the emergency of the COVID-19 pandemic [2]. Hence, this article presents the complete dataset, and especially the method that was used to collect, analyze, and visualize the data.

The dataset of 14,952 news articles can be used for future studies regarding COVID-19 in Vietnam. Furthermore, the method offers a low-cost option for researchers who are under financial constraints to conduct similar studies [21,22]. Meanwhile, the method's strengths lie in its ability to gather a large amount of data and flexibly filter and analyze it in a short time. Speed is an advantage in learning about the impact of important events. Moreover, for a different research purpose, the method can help flexibly filter the data both manually and automatically.

Thus far, the method has contributed to two publications on the COVID-19 pandemic and corporate social responsibility in Vietnam [2,3]. In different circumstances, the method had proven its strengths, while still leaving room for improvements. As previously mentioned, one publication [2] was done in the urgent atmosphere of the early days of the COVID-19 pandemic. Thus, the methods allowed the authors to collect a large amount of news media data and to quickly analyze the data. Meanwhile, the study on CSR missions in Vietnam was done by manually sorting through the news articles and assigning their characteristics to research objects [3].

The article provided a validated dataset and its method for collecting, cleaning, and analyzing the data. While the dataset can be used for future analysis of COVID-19 in Vietnam, the method provides a cheap but effective option for conducting research about news media. The tool and method are particularly suitable and useful in acquiring information from sources that are difficult to validate and check. It is thus intended by our

research team that the approach will be used in our near-future research program dealing with the world's most important issues, such as climate change, environmental protection, healthcare, and regulation compliance behaviors in society [3,5,23–25]. The data extracted by this method, upon appropriate categorization, can work well with the *bayesvl* R package, also developed by AISDL [26–28].

**Author Contributions:** Q.-H.V. and M.-T.H. conceptualized and design the study; Q.-H.V. and V.-P.L. design the software/system; M.-T.H., T.-H.T.N., M.-H.N. and T.-T.L. wrote and revised the manuscript; Q.-H.V. supervised and administered the study. All authors have read and agreed to the published version of the manuscript.

**Funding:** This research received no external funding.

**Institutional Review Board Statement:** Not applicable.

**Informed Consent Statement:** Not applicable.

**Data Availability Statement:** The dataset is available on Zenodo: https://zenodo.org/record/4738602 (accessed on 25 June 2021); doi:10.5281/zenodo.4738602.

**Acknowledgments:** We thank the research staffs at AISDL for their assistance and prior studies that enable the writing of this method article, namely: Dam Thu Ha, Vuong Thu Trang, Ho Manh Tung, Nguyen To Hong Kong.

**Conflicts of Interest:** The authors declare no conflict of interest.

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
