# Peer review of "An AI-Enabled Approach in Analyzing Media Data: An Example from Data on COVID-19 News Coverage in Vietnam"

_data_

Round 1
Reviewer 1 Report
Well drafted and interesting introduction to the methodology implementation for extracting and analyzing social media data.
I don't have particular comments
Author Response
Dear Reviewer 1,
We have provided point-to-point responses to the comments of reviewers in our revised version. Please note that in the revised manuscript, the parts that are highlighted in yellow denote corrections to the old text, while the parts highlighted in green have been written anew. Below are our modifications and answers to the editor’s and reviewers’ comments (in bold font).
Well drafted and interesting introduction to the methodology implementation for extracting and analyzing social media data.
I don't have particular comments
Thank you for your encouraging comments.
Once again, we appreciate the hard work and time that you have spent on this manuscript. We hope that the revised paper has met your requirements.
Please accept our sincere thanks for your great contributions to the improvement of the overall advancement of sciences in the world.
Shall you have further comments, we look forward to hearing from you.
Yours sincerely,
On behalf of all the authors,
Reviewer 2 Report
Authors propose a framework for acquiring and analyzing textual data coming from news.
The novelty is limited since they put together Python libraries for web scraping, text pre-processing and data analysis.
Firstly I suggest to depict a schema of the proposed tool. Then, authors must highlight which are the main contributions of this article, and how the proposed tool is novel if compared with other existing tools. Indeed a review of the state-of-the-art is missing.
All the images are blurry, please use high quality images.
Please remove from pages 5 and 6 the tables on the left. Logic propositions are auto-explicative.
Authors refer to articles 2 and 3. Please highlight which are the main differences among these articles and this one.
Visualization techniques should be better described to understand how to interpret the results.
Author Response
Dear Reviewer 2
We have provided point-to-point responses to the comments of reviewers in our revised version. Please note that in the revised manuscript, the parts that are highlighted in yellow denote corrections to the old text, while the parts highlighted in green have been written anew. Below are our modifications and answers to the editor’s and reviewers’ comments (in bold font).
Authors propose a framework for acquiring and analyzing textual data coming from news.
The novelty is limited since they put together Python libraries for web scraping, text pre-processing and data analysis.
Firstly I suggest to depict a schema of the proposed tool. Then, authors must highlight which are the main contributions of this article, and how the proposed tool is novel if compared with other existing tools. Indeed a review of the state-of-the-art is missing.
We have revised the article heavily according to your suggestion. For instance, in the Summary section, a review of the state-of-the-art was provided:
Due to this reason, research switched to tools which could automatically scan news outlets and collect news articles to save time and cost [7]. For example, Brandy and Diakopoulos created Python programs to curate stories and the “share” links of each news article before storing them in a csv. file. They did not collect the content of these articles because the purpose of their study was only to compare the degree of personalization between human-curated and algorithmically curated sections [8]. There are papers which manually collected news before automatically analyzed the content. A study conducted in China collected news in the China National Knowledge Infrastructure (CNKI) database using a set of keywords related to COVID-19 and tourism retrieved 794 articles published in a month before yielding only 499 articles for analysis [9]. This task would be burdensome in the case of extending the period of publishing, number of news outlets and websites, where we could obtain thousands of articles. This study did use an automated program to tokenize news articles and extract the top keywords. However, no detail concerning how accurately the program used in this study processed the language was provided. On this matter, despite there are toolkits which were successfully applied to English documents [10], studies about Vietnamese documents had a concern whether techniques which originally used on English documents could be applied to Vietnamese documents because these languages have many differences in processing [11, 12, 13]. Researchers proposed using Bag of Words with keyword extraction and Neural Network to solve the problem related to Vietnamese news classification [11].
During the outbreak of COVID-19 in Vietnam in March 2020, our research team has decided to study the COVID-prevention policy from the government, the reaction from the media, and the scientific community. Due to the number of websites and the long period featured in the study, the number of articles was expected to be too much to obtain manually. Furthermore, the concern regarding to the ability of machine to process the content of articles was prevailing. With this purpose, we needed a tool which can: (1) scan Vietnamese news outlets and official government websites to identify news articles with a set of keywords, download their raw HTML data and extract unrelated content; (2) remove duplicated articles (similarity over 90%); (3) process Vietnamese effectively. However, there was not a tool that is suitable for acquiring data satisfying our needs. Available tools can brilliantly perform the first and second steps but not the third, due to the difficulties to split Vietnamese words and phrases.
Difficulties to process Vietnamese words and phrases correctly with machines have been well documented [11-14]. While tokens of languages such as English and Russian are words, Vietnamese has sub-syllables, syllables and words [15] so using homogeneous unit systems could lead to incorrect splitting and disastrous misunderstanding. Furthermore, Vietnamese language is complex, so it is easily to split words and phrases in the wrong way. Due to which, tokenizing Vietnamese phrases should be performed carefully based on the context to achieve the best results. In addition, Vietnamese has words which are written in the same way but have different meaning. This could lead to misleading analysis. For example, the word “cÆ¡ quan” could mean “body organs” or “a government agency”, depending on the context. When tool lacks the ability to extract the contextual information, which is usually hidden and not extractable [16, 17], the tool could misunderstand the meaning of a word even when a sentence has been split correctly, and this might miscount the frequency that the word appears, which lead to wrong analysis. When the COVID-19 pandemic influenced a broad range of fields, there might be a variation of meaning across different context. Thus, there are words required careful definition before being used in other fields to avoid misunderstanding [18]. For these reasons, we needed to customize our programs, which is a homemade AI-enabled software using the Python codes, to collect news from online sources and process Vietnamese better. Effective use of AI module was suggested to be the solution to problems of information retrieval and analysis [19].
All the images are blurry, please use high quality images.
We have replaced all the images with higher quality ones. Thank you for pointing out this obvious shortcoming.
Please remove from pages 5 and 6 the tables on the left. Logic propositions are auto-explicative.
We have removed the tables accordingly. Thank you for your suggestion.
Authors refer to articles 2 and 3. Please highlight which are the main differences among these articles and this one.
This article is a data descriptor, which focuses on describing the dataset and the methodology to collect and analyze the data. According to the Instruction for Authors page, a data descriptor is (https://www.mdpi.com/journal/data/instructions):
Data Descriptors: containing a description of a dataset, including methods used for collecting or producing the data, where the dataset may be found, and information about its use.
Thus, this article is entirely different from article [2] and [3]. The articles [2] and [3] provided in-depth results and analysis, while this article focused entirely on a particular dataset and its methods used for collecting and analyzing the data.
Visualization techniques should be better described to understand how to interpret the results.
We have provided more detail on the visualization: We use the d3.js library (https://d3js.org) to visualize the research results. The crawled news articles were stored according to its published time. Therefore, we were able to analyze the data in different time period.
Once again, we appreciate the hard work and time that you have spent on this manuscript. We hope that the revised paper has met your requirements.
Please accept our sincere thanks for your great contributions to the improvement of the overall advancement of sciences in the world.
Shall you have further comments, we look forward to hearing from you.
Yours sincerely,
On behalf of all the authors,

Reviewer 3 Report
This paper presents an AI-enabled web crawling approach to extract analyze social media data.
Focused on COVID-19, the authors extracted a dataset of more than 14,000 news articles from Vietnamese newspapers to provide a comprehensive picture of how Vietnam responds to the pandemic.
The article is well-structured and presents interesting information.
The datasets are also made publicly available, which is a valuable aspect of the work.
My main concerns are the following:
- What are the merit and special features of the AI-powered crawling approach?
There is no section where this aspect is addressed.
I would expect an analysis where the authors cite other studies where the authors highlight the advantages of their approach.
Five references for a journal publication is not acceptable.
In addition, emphasize the role of the AI component, also in comparison with other studies (See for example DOI: 10.3390/biology9120453, 10.1109/NICS.2018.8606890, 10.1109/IC-AIAI48757.2019.00023, 10.20517/jsss.2020.15, 10.1109/REW.2017.20)
- Figure quality is very low. When zoomed, figures become grainy and unreadable.
I recommend the authors to prepare a revised version of the manuscript.
Author Response
Dear Reviewer 3,
We have provided point-to-point responses to the comments of reviewers in our revised version. Please note that in the revised manuscript, the parts that are highlighted in yellow denote corrections to the old text, while the parts highlighted in green have been written anew. Below are our modifications and answers to the editor’s and reviewers’ comments (in bold font).
This paper presents an AI-enabled web crawling approach to extract analyze social media data.
Focused on COVID-19, the authors extracted a dataset of more than 14,000 news articles from Vietnamese newspapers to provide a comprehensive picture of how Vietnam responds to the pandemic.
The article is well-structured and presents interesting information.
The datasets are also made publicly available, which is a valuable aspect of the work.
Thank you for your encouraging comments
My main concerns are the following:
- What are the merit and special features of the AI-powered crawling approach?
There is no section where this aspect is addressed.
We have provided a Usage Notes section to provide potential applications of this methods. Moreover, drawing from the literature review, special features and contributions to the development of the methods were also discussed.
The dataset provided more than 14,000 news articles about COVID-19, and it was quickly analyzed in just a week to meet the emergency of the COVID-19 pandemic [2]. Hence, this article presented the complete dataset, and especially the method that was used to collect, analyze, and visualize the data.
The dataset of 14,952 news articles can be used for future studies regarding COVID-19 in Vietnam. Furthermore, the method offers a low-cost option for researchers who are under financial constraints to conduct similar studies [21,22]. Meanwhile, the method’s strengths lie in the ability to gather a large amount of data and flexibly filter and analyze it in a short time. The speed is an advantage in learning about the impact of important events. Moreover, in a different research purpose, the method can help flexibly filter the data both manually and automatically.
Thus far, the method has contributed to two publications on the COVID-19 pandemic and the corporate social responsibility in Vietnam [2, 3]. In different circumstances, the method had proven its strengths, while still left room for improvements. As previously mentioned, the publication [2] was done in an urgent atmosphere of the early days of the COVID-19 pandemic. Thus, the methods allowed the authors to collect a large amount of news media data and quickly analyzed the data. Meanwhile, the study on CSR missions in Vietnam was done by manually sorting through the news article and assigning characteristics to research objects [3].
The article provided a validated dataset and its method for collecting, cleaning and analyzing the data. While the dataset can be used for future analysis of COVID-19 in Vietnam, the method provides a cheap but effective option for conducting research about news media. The tool and method are particularly suitable and useful in acquiring information from sources that are difficult to validate and check. It is, thus, intended by our research team that the approach will be used in our near-future research program dealing with the world’s most important issues such as climate change, environment protection and regulations compliance behaviors in society [3, 5, 23]. The data extracted by this method, upon appropriate categorization, can work well with the bayesvl R package, also developed by AISDL [24-26].
I would expect an analysis where the authors cite other studies where the authors highlight the advantages of their approach. Five references for a journal publication is not acceptable.
In addition, emphasize the role of the AI component, also in comparison with other studies (See for example DOI: 10.3390/biology9120453, 10.1109/NICS.2018.8606890, 10.1109/IC-AIAI48757.2019.00023, 10.20517/jsss.2020.15, 10.1109/REW.2017.20)
Thank you for your suggestion. We have expanded the Summary section to provide a literature review for the method. Your suggested papers were useful and they were included as well:
Due to this reason, research switched to tools which could automatically scan news outlets and collect news articles to save time and cost [7]. For example, Brandy and Diakopoulos created Python programs to curate stories and the “share” links of each news article before storing them in a csv. file. They did not collect the content of these articles because the purpose of their study was only to compare the degree of personalization between human-curated and algorithmically curated sections [8]. There are papers which manually collected news before automatically analyzed the content. A study conducted in China collected news in the China National Knowledge Infrastructure (CNKI) database using a set of keywords related to COVID-19 and tourism retrieved 794 articles published in a month before yielding only 499 articles for analysis [9]. This task would be burdensome in the case of extending the period of publishing, number of news outlets and websites, where we could obtain thousands of articles. This study did use an automated program to tokenize news articles and extract the top keywords. However, no detail concerning how accurately the program used in this study processed the language was provided. On this matter, despite there are toolkits which were successfully applied to English documents [10], studies about Vietnamese documents had a concern whether techniques which originally used on English documents could be applied to Vietnamese documents because these languages have many differences in processing [11, 12, 13]. Researchers proposed using Bag of Words with keyword extraction and Neural Network to solve the problem related to Vietnamese news classification [11].
During the outbreak of COVID-19 in Vietnam in March 2020, our research team has decided to study the COVID-prevention policy from the government, the reaction from the media, and the scientific community. Due to the number of websites and the long period featured in the study, the number of articles was expected to be too much to obtain manually. Furthermore, the concern regarding to the ability of machine to process the content of articles was prevailing. With this purpose, we needed a tool which can: (1) scan Vietnamese news outlets and official government websites to identify news articles with a set of keywords, download their raw HTML data and extract unrelated content; (2) remove duplicated articles (similarity over 90%); (3) process Vietnamese effectively. However, there was not a tool that is suitable for acquiring data satisfying our needs. Available tools can brilliantly perform the first and second steps but not the third, due to the difficulties to split Vietnamese words and phrases.
Difficulties to process Vietnamese words and phrases correctly with machines have been well documented [11-14]. While tokens of languages such as English and Russian are words, Vietnamese has sub-syllables, syllables and words [15] so using homogeneous unit systems could lead to incorrect splitting and disastrous misunderstanding. Furthermore, Vietnamese language is complex, so it is easily to split words and phrases in the wrong way. Due to which, tokenizing Vietnamese phrases should be performed carefully based on the context to achieve the best results. In addition, Vietnamese has words which are written in the same way but have different meaning. This could lead to misleading analysis. For example, the word “cÆ¡ quan” could mean “body organs” or “a government agency”, depending on the context. When tool lacks the ability to extract the contextual information, which is usually hidden and not extractable [16, 17], the tool could misunderstand the meaning of a word even when a sentence has been split correctly, and this might miscount the frequency that the word appears, which lead to wrong analysis. When the COVID-19 pandemic influenced a broad range of fields, there might be a variation of meaning across different context. Thus, there are words required careful definition before being used in other fields to avoid misunderstanding [18]. For these reasons, we needed to customize our programs, which is a homemade AI-enabled software using the Python codes, to collect news from online sources and process Vietnamese better. Effective use of AI module was suggested to be the solution to problems of information retrieval and analysis [19].
- Figure quality is very low. When zoomed, figures become grainy and unreadable.
We have updated the Figure and provided a high-quality version to the journal for later stage of the production.
I recommend the authors to prepare a revised version of the manuscript.
Once again, we appreciate the hard work and time that you have spent on this manuscript. We hope that the revised paper has met your requirements.
Please accept our sincere thanks for your great contributions to the improvement of the overall advancement of sciences in the world.
Shall you have further comments, we look forward to hearing from you.
Yours sincerely,
On behalf of all the authors,

Round 2
Reviewer 2 Report
Authors have improved the article according with the reviewers' suggestions.
Reviewer 3 Report
The authors provided an improved version of the paper.
In particular, they:
- Emphasized the merit and special features of the AI-powered crawling approach
- Increased the number of references to related studies
- Increased the quality of the figures.
For these reasons, I recommend the paper being accepted for publication.